# Bacterial Colony Phenotyping with Hyperspectral Elastic Light Scattering Patterns

**DOI:** 10.3390/s23073485

**Published:** 2023-03-27

**Authors:** Iyll-Joon Doh, Diana Vanessa Sarria Zuniga, Sungho Shin, Robert E. Pruitt, Bartek Rajwa, J. Paul Robinson, Euiwon Bae

**Affiliations:** 1Applied Optics Laboratory, School of Mechanical Engineering, Purdue University, West Lafayette, IN 47907, USA; 2Department of Botany and Plant Pathology, Purdue University, West Lafayette, IN 47907, USA; 3Department of Basic Medical Sciences, College of Veterinary Medicine, Purdue University, West Lafayette, IN 47907, USA; 4Bindley Bioscience Center, Purdue University, West Lafayette, IN 47907, USA; 5Weldon School of Biomedical Engineering, Purdue University, West Lafayette, IN 47907, USA

**Keywords:** bacterial identification, elastic light scattering, bacterial colony phenotyping, hyperspectral imaging, light diffraction, optical sensing, supercontinuum laser

## Abstract

The elastic light-scatter (ELS) technique, which detects and discriminates microbial organisms based on the light-scatter pattern of their colonies, has demonstrated excellent classification accuracy in pathogen screening tasks. The implementation of the multispectral approach has brought further advantages and motivated the design and validation of a hyperspectral elastic light-scatter phenotyping instrument (HESPI). The newly developed instrument consists of a supercontinuum (SC) laser and an acousto-optic tunable filter (AOTF). The use of these two components provided a broad spectrum of excitation light and a rapid selection of the wavelength of interest, which enables the collection of multiple spectral patterns for each colony instead of relying on single band analysis. The performance was validated by classifying microflora of green-leafed vegetables using the hyperspectral ELS patterns of the bacterial colonies. The accuracy ranged from 88.7% to 93.2% when the classification was performed with the scattering pattern created at a wavelength within the 473–709 nm region. When all of the hyperspectral ELS patterns were used, owing to the vastly increased size of the data, feature reduction and selection algorithms were utilized to enhance the robustness and ultimately lessen the complexity of the data collection. A new classification model with the feature reduction process improved the overall classification rate to 95.9%.

## 1. Introduction

Foodborne illnesses are caused mainly by the consumption of food that is contaminated with pathogenic bacteria, viruses, parasites, or chemicals [1]. Early detection of these contaminants is critical in reducing morbidity and preventing costly recalls of contaminated foods during an outbreak. Therefore, faster, easier-to-use, and more reliable detection and organism identification are crucial in the food industry and other related areas. Optical metrology for bacterial identification has gained interest due to its convenience, noninvasiveness, lack of contact with the measured material, and speed [2]. Optical scattering and diffraction pattern-based identification methods such as the elastic light scattering (ELS) technique [3,4,5] and Bacterial Identification System by Light Diffraction (BISLD) [6] are shown to be rapid, nondestructive, and label-free, demonstrating potential applicability in the clinical and food safety sectors. The core technology behind these two optical techniques is the same and involves identifying bacterial species based on the forward scattering and diffraction patterns of their colonies. A volumetric photon-cell interaction explains the fundamental mechanism underpinning pattern creation. The interplay between light, individual cells, and extracellular material forms a highly unique 2D diffraction pattern that provides information about individual cell features and the aggregate morphological qualities of a colony [3].

The previously demonstrated pattern-generation methodology utilized a single-wavelength 635 nm light source to test a variety of microbial organisms, with the performance being reported in several publications [7,8]. However, owing to the morphological similarity of colonies formed by phylogenetically close microorganisms, this established technique performed poorly when attempting to identify them. Several techniques, such as multiwavelength and multichannel systems, have been developed to address this issue and enhance classification performance [9,10,11]. When the multispectral ELS system incorporating 405, 635, and 904 nm diode lasers was tested with seven distinct *E. coli* spp. serovars, it demonstrated an improvement over the single-wavelength technique [9]. The features were extracted from the multispectral patterns, and the random forest (RF) algorithm was adopted to select highly predictive features for the classification model. This successful early implementation of a multispectral system accompanied by machine learning motivated the creation of a new ELS system that utilizes more wavelengths and relies on hyperspectral ELS patterns for improved classification.

The major differences between the hyperspectral and multispectral approaches are the number and resolution of the spectral bands employed in the measurement. Multispectral systems collect signals from a few discrete spectral bands, whereas the hyperspectral approaches utilize a large number of bands approximating a continuous spectrum [12]. Therefore, hyperspectral techniques can provide more in-depth details and generate more precise spectral fingerprints than multispectral techniques [13]. Hyperspectral imaging techniques thus can leverage the benefit of traditional digital imaging, computer vision, and spectroscopy to process both spatial and spectral information [14]. Since hyperspectral imaging requires more spectral bands than multispectral imaging, highly sophisticated hardware and software is needed. This raises the cost of data acquisition and processing [15].

A hyperspectral system typically requires a broadband source and wavelength modulator that efficiently isolates the light at a single wavelength so that the imaging sensor can capture the spatial information of an object in a variety of wavelengths [16]. Since the light source must be coherent in the ELS system to create the diffraction pattern, a supercontinuum (SC) laser is an attractive option for this application due to its broad spectrum and high coherence properties [17]. SC laser is a broadband light source generated when an optical pulse propagates through an extremely nonlinear medium. For the wavelength modulator, an acousto-optic tunable filter (AOTF) is popular tool that utilizes acousto-optical interaction through birefringent medium to modulate the wavelength of light. The anisotropic Bragg diffraction of broadband light triggered from the acoustic signal passing through the birefringent medium provides excellent optical and technical characteristics, including high spectral and spatial resolution, a broad tuning range, no moving parts, and rapid wavelength tuning speed [18]. Therefore, AOTF is widely utilized in biomedical applications that require wavelength selection such as hyperspectral imaging [19], flow cytometry [20], and fluorescence and confocal microscopy [21,22]. The SC lasers are also often paired with acousto-optic tunable filters (AOTFs) in various applications to precisely control the interrogating wavelength [23,24]. By integrating these components with the ELS system, it is possible to generate the hyperspectral pattern and perform simultaneous measurement. The ELS technique differs from the imaging or microscopic technologies mentioned earlier in that it classifies bacterial colonies based on their Fresnel diffraction and scattering patterns. This technique directly measures the spatial intensity of the pattern without using any optics. As a result, it is not subject to any imaging aberrations caused by factors such as chromatic aberration and acousto-optic interaction, making it free from the need for calibrations to correct such distortions. Along with the spectral resolution improvement, the hyperspectral sensing approach also increases the data dimensionality. This can bring additional challenges because of the increased number of images [25]. Thus, feature or variable selection approaches are often employed to decrease the possibility of overfitting and lower the computational complexity while preserving or improving classification accuracy [26]. Feature extraction and selection are crucial steps in a pattern or image classification. Their importance is demonstrated in various applications, including PM2.5 feature recognition and air pollution control [27,28,29].

Herein, we report the development and validation of a hyperspectral ELS phenotyping instrument (HESPI). The novel device utilized an SC laser as a broadband light source and an AOTF to select the wavelength of interest for the ELS pattern generation. The classification ability of the hyperspectral ELS system was evaluated using eight different bacteria species of *Arthrobacter* sp., *Curtobacterium* sp., *Massilia* sp., and *Microbacterium* sp., which are a part of the natural microflora found in a green leaf vegetable such as lettuce. With the newly developed instrument, a total of 70 scattering patterns were collected within the wavelength range 473–709 nm for each colony, and descriptive features were extracted from the patterns for classification purposes. As a result of the increased amount of data, a machine learning technique was required to process the variables and construct the classification model.

## 2. Materials and Methods

### 2.1. Instrumentation Design

HESPI consisted of four major parts: a white light source, a wavelength modulator, an image sensor, and a sequence controller. The schematic in Figure 1 illustrates the path of a laser beam (solid arrows) originating from the SC laser and the sequence of the system control (dotted arrows). For the white light source, an SC laser (SC-5, YSL Photonics, Wuhan, China) whose spectrum covers 450–2400 nm was utilized. The laser provided a collimated beam with a diameter of about 2 mm at 633 nm. The repetition rate of the pulse was fixed to 5 MHz, resulting in a total power of approximately 800 mW. The overall power of the laser was controlled from the PC via a USB connector. A 45° cold mirror (FM03, Thorlabs, Inc., Newton, NJ, USA) was placed right after the laser to split the beam into VIS and IR. The transmitted IR portion was disposed of in a beam dump, and only the VIS portion was applied to create the ELS pattern. For the wavelength modulator, an AOTF crystal block obtained from a Brimrose AOTF Microscope video adaptor (MIM-200, Brimrose Corp., Sparks Glencoe, MD, USA) was utilized to select a specific wavelength of interest. The filter was connected to an RF driver (VFI-139-90-SPS-A-C2, Brimrose Corp.), which consists of a MHz frequency generator with an amplitude modulator to control the wavelength and amplitude of the diffracted beam. The frequency and the percent amplitude were set using ASCII commands sent through a serial port. The driver generated an acoustic signal from 95 to 180 MHz, equivalent to 454–749 nm in wavelength. The spectral resolution of the AOTF was 2.4 nm at 488 nm and 5.5 nm at 633 nm. At the exit of the AOTF, the beam was split into unfiltered (0th order) and filtered (1st order) beams by acoustic signal causing the diffraction. The exit angle of the diffracted beam varied by the wavelength, and to compensate for the various diffraction angles, two identical biconvex lenses (LB1723, Thorlabs Inc.) whose diameter and focal lengths were 2 inches and 60 mm were positioned after the AOTF. At the target, the spot size of the laser beam was about 1.5–2 mm to fully cover the 1 mm diameter bacterial colony, and the power of the laser beam was also varied by wavelength because of the spectral efficiency of the SC laser. The measured beam power in μW, depending on the wavelength, is presented in the Appendix A. A monochromatic CMOS sensor (PL-B741, Pixelink Corp., Gloucester, ON, Canada) whose resolution was 1280 × 1024 pixels with a 6.7 μm unit pixel size was placed 40 mm below a Petri dish and captured the scattering pattern with 0.1 s of exposure time and 0 gain. The captured scattering patterns were then stored on a PC for further analysis. The total acquisition time was less than 30 s for scanning the colony for 70 times to capture 70 spectral patterns. The pictures of the actual HESPI setup are provided in the Appendix A.

### 2.2. Sample Preparation

Samples representing a range of bacterial species, including *Arthrobacter* sp., *Curtobacterium* sp., *Massilia* sp., and *Microbacterium* sp., were cultured to test the classification performance of the hyperspectral system. These organisms represent natural microflora found in green leafy foods such as lettuce. They were chosen based on the classification performance of the conventional single-wavelength approach, which had previously demonstrated poor success in separating these eight bacterial species (see Appendix A). The selected bacteria were 128—*Massilia*, 284—*Microbacterium*, 410—*Microbacterium*, 441—*Microbacterium*, 510—*Arthrobacter*, 526—*Arthrobacter*, 536—*Curtobacterium*, and 586—*Curtobacterium*. The code numbers before the genera name represent different strains. Bacterial samples were prepared by culturing on plate count agar (PCA). Stock cultures were prepared from frozen stocks stored at −80 °C, streaked on PCA, and incubated at 30 °C until colonies could be identified visually. For each species, one colony was randomly selected, picked from the streaked plate, and diluted in 4 mL buffer solution (PBS) followed by three serial dilutions of 1:40 in PBS. A 50 μL aliquot of the last dilution tube was spread on PCA using a sterile hockey stick. The plates were incubated at 30 °C until the diameters of the colonies were between 800 and 1000 μm.

### 2.3. Pattern Image Preprocessing

In the region of shorter wavelengths, 450–550 nm, the spectral inefficiency of the SC laser led to a decrease in laser beam strength, which negatively affected the quality of the recorded scattering patterns. For this reason, the scattering patterns generated at this wavelength range had relatively lower contrast, and the diffraction rings were hardly visible. Therefore, image preprocessing was performed using a MATLAB procedure to increase the contrast in the scattering patterns. For each pattern, the intensity values were adjusted based on a reference pattern whose average intensity was the highest among 70 hyperspectral data sets. This operation leveled the intensity of the pattern images and increased the contrast. Details and an example of image preprocessing are presented in the Appendix A.

### 2.4. Feature Extraction

As phenotypic characteristics of the bacterial colony lie in the scattering patterns, extracting appropriate descriptive features from the patterns is the key to successful bacterial classification. Owing to the specific physical properties of the analyzed images, features based on complex moment invariants such as Zernike and pseudo-Zernike moments are preferred over simple geometric descriptors such as area, perimeter, roundness, and so forth [30]. Zernike moment invariance is a particularly attractive feature as the readout is not affected by the rotation of the scatter pattern, and the magnitude remains unchanged. Therefore, this approach works well with circular and symmetric patterns [30]. For this study, we employed the magnitudes of pseudo-Zernike moment (PZM) as invariant features. They are computed using pseudo-Zernike polynomials, which are a set of orthogonal polynomials with similar properties to Zernike polynomials but are recognized as being less sensitive to noise than conventional Zernike moments [31].

### 2.5. Classification and Feature Interpretation

The classification model was constructed using the extracted feature set. A separate testing set containing a subset of the data was used to evaluate the constructed classifier. The ratio between the training and testing sets was 7:3. This study applied two classification strategies: the use of trained classifiers specific to each wavelength and the use of a single classifier trained on all wavelengths as a single set. The number of variables in the feature set varied depending on the classification methods. For a single-wavelength classifier, we utilized the linear support vector machine (SVM) model. When using all 70 wavelengths, it was necessary to use a feature selection technique to minimize the dimensionality of the feature space by deleting strongly correlated features that severely impact classification accuracy. The excessive number of variables in the feature set can increase the computational cost of the training, and more importantly, the increased dimensionality can cause an overfitting [32]. Therefore, two steps of feature selection were implemented in this study: univariate and multivariate feature selections.

Univariate feature selection evaluates every single feature separately, whereas multivariate feature selection evaluates the entire feature subset [33]. A linear correlation-based filter mechanism, the analysis of variance (ANOVA), was used for the univariate step. Hence, each variable in the feature set was evaluated by building a linear model and inspecting the observed effect size (η^2^), representing the proportion of explained variance [34]. The features associated with models that show smaller effect sizes were filtered out for the set. The remaining features were then processed further by a multivariate feature selection process.

The multivariate feature selection was performed with the help of multinomial logistic regression with elastic net regularization (ENET) [35]. Combining ℓ_1_ (LASSO) and ℓ_2_ (ridge) regularization, this classification method incorporates feature selection, allowing for simultaneous training and variable selection [36]. The procedure was implemented in Python with open-source libraries from Scikit-learn [37]. The grid search across the space of parameters was used to determine the strength of regularization and the relative contribution of ℓ_1_ and ℓ_2_ penalties.

The performance of classifier was presented by the cross-validation matrix. Based on the CV matrix, five statistical parameters were calculated: accuracy, sensitivity, specificity, positive predictive value (PPV), and negative predictive value (NPV):(1)Accuracy=(TP+TN)/(TP+FP+TN+FN)
(2)Sensitivity=TP/(TP+FN)
(3)Specificity=TN/(FP+TN)
(4)PPV=TP/(TP+FP)
(5)NPV=TN/(TN+FN)
where *TP*, *TN*, *FP*, and *FN* stand for true positive, true negative, false positive, and false negative, respectively.

After the classification was performed, the model was interrogated to find the features that were contributing the most to its predictive ability. The feature importance is directly related to the absolute value of the corresponding model coefficients. However, we also used an independent feature ranking via a local interpretable model-agnostic explanation (LIME) algorithm. LIME is a novel technique that explains the prediction performed by a classifier by constructing an interpretable local model around the prediction [38]. LIME modifies an instance by adjusting the feature values and evaluates the output’s response. This method provides local interpretability by describing the contribution of each feature to the prediction. The feature rankings from LIME and the ENET were then cross-checked for verification. Figure 2 is a flowchart depicting all the described classification steps for hyperspectral light-scatter patterns, including feature selection and interpretation procedures.

## 3. Results

### 3.1. Hyperspectral ELS Patterns

Hyperspectral ELS patterns of the bacteria samples were collected using the HESPI system. For each organism, 50 colonies were randomly selected, and 70 hyperspectral patterns were collected per colony. The wavelength of the laser was controlled by the acoustic frequency through the RF driver, ranging from 101 to 170 MHz, which provided a spectral range of 473–709 nm. Figure 3 shows the representative images of hyperspectral patterns for the sample bacteria colonies. The rows represent the bacteria species, whereas the columns represent different wavelengths. The pattern images are grouped in red boxes according to their genus type, indicating that there are multiple species sharing the same genus. The successful classification of this sample group brings the hyperspectral approach closer to being a species-level classification tool. From the visual inspection, it is apparent that seven out of eight bacteria produced concentric ring-type patterns. The scattering patterns of 526—*Arthrobacter* in Figure 3F were distinct because the pattern was too large to fit within the active area of the imaging sensor. Consequently, only the center portion of the patterns is shown in the displayed images. Additionally, it was observed that the shapes of the patterns changed as the wavelength changed. More specifically, the number, thickness, and distance between the diffraction rings changed. Previous studies on the wavelength-dependent ELS pattern characteristics predicted and explicitly explained this phenomenon. Our previous work simulated the spectral effect on the shape of ELS patterns using a scalar diffraction model. The simulation revealed that the ring gap and ring width were proportional to the incident wavelength, while the number of rings and half diffraction angle were inversely proportional. This phenomenon was experimentally verified by measuring the change in the scattering patterns of *S. aureus* colonies with respect to the incident wavelength [7].

### 3.2. Hyperspectral ELS Pattern Classification

The patterns of hyperspectral light scattering were initially analyzed separately according to their respective wavelengths. PZMs were extracted from each pattern using the 10th-order pseudo-Zernike polynomials, which yielded the most accurate classification results demonstrated after an iterative search. The detailed explanation and result of the order search are provided in the Appendix A. Figure 4 illustrates the classification performance of each wavelength as a heatmap to visually represent the positive predictive value (PPV) for each organism. For every wavelength, the classifier was trained using linear SVM with 10 × 2 cross-validation (CV). The overall average PPV was 92.7%, and among the eight bacteria, 284—*Microbacterium* resulted in the lowest average PPV with 85.7%, while 526—*Arthrobacter* achieved the highest average PPV with 99.9%. In general, 284—*Microbacterium*, 410—*Microbacterium*, and 441—*Microbacterium* resulted in PPVs below the average because of the low classification efficiency in the 600–700 nm region. In addition, each organism was associated with a single wavelength that yielded the best classification performance. For example, the 128—*Massilia* classifier had the highest PPV around the 532 nm region, which implies that 128—*Massilia* is likely to be differentiable when the scattering patterns are created with the 532 nm light source. This indicates that using several wavelengths would be more effective for distinguishing between species than one wavelength. The second study used all the features from the entire wavelength range for classification. A new feature set was created by combining all the variables. As a result of combining all 70 wavelengths, the number of features was 70 times greater than with the previous classification method. The new linear SVM yielded an average classification accuracy of approximately 96.4%, indicating that incorporating the entire wavelength range improved classification performance over single-wavelength classification. A table with individual values is provided in the Appendix A.

### 3.3. Feature Selection and Classification

When integrating the entire spectral range into the classification model, the total number of variables for each observation was 9450 (for the PZM order set to 15). There were 70 wavelengths, and each wavelength contained a vector of 135 invariant features. Owing to a large number of features relative to the number of observations, there was a risk of the classification model overfitting. Consequently, a feature selection procedure was employed during classification. For the univariate feature selection, a series of linear models were built, and the effect size demonstrated in each model was determined in order to rank them. The top 300 out of 9450 features were chosen based on the effect size, and the feature number was determined by classification score, which was tested with feature numbers ranging from 10 to 500 (see Appendix A). It was discovered that the number of selected features was the same for each wavelength. Before the filtration, each wavelength had a vector of 135 features, and after the filtration, only the 1st, 2nd, 3rd, and 10th index of features representing PZM invariants remained in every wavelength.

The filtered features were subsequently utilized in the multivariate feature selection step. The ENET logistic regression model was used with the tuned hyperparameters. The search was repeated 10 times, and the result showed that the best classification was achieved when the ℓ_1_ contribution was 100%. This means that the ENET regularization employed only the LASSO penalty. Figure 5 depicts the classification result as five statistical parameters calculated based on the CV matrix. Despite the reduced number of features, the performance was promising, as the averages of the parameters were greater than 90%. In terms of PPV, the results from both the ENET and the linear SVM demonstrated that the classification using all of the wavelengths resulted in improved performance over the single wavelength method. The average PPV improved from 93.1 ± 2.1% to 96.2 ± 3.0% for ENET and 96.4 ± 1.78% for the linear SVM, as all wavelength bands were utilized to build the classifier. The comparison shown in Table 1 demonstrates that the PPVs of most organisms were higher when the entire feature set was used for classification. This result indicated that hyperspectral ELS data produce a more accurate classification model than single wavelength patterns.

The comparison between the ENET result and single-wavelength classification using SVM is provided in Figure 6. This comparison demonstrates, once again, the improvements to classification by hyperspectral application in the ELS method. The horizontal red dotted line represents the average classification score of the ENET-based system, whereas the bars represent the classification performance at every separate wavelength. Although some wavelengths between 520 and 540 nm used with the SVM classifier produced performance comparable to that of the hyperspectral ENET, the majority of wavelengths resulted in lower classification scores. The wavelengths represented by the bars highlighted in blue are 488, 532, 561, 635, 650, and 670 nm, as they are easily available in the commercial diode lasers. This shows that the hyperspectral approach with a white-light laser can easily create scattering patterns in wavelengths that are not available in conventional diode laser format.

### 3.4. Feature Interpretation

LIME provides a method for explaining the mechanism underlying the classification model’s predictions. Since a multinomial logistic regression model was used as the classifier for this investigation, each observation was assigned to the class with the highest probability. LIME enabled the creation of an ordered list of features based on their contribution to the classifier’s judgment. For each observation, the top 25 classification-relevant characteristics were evaluated. The contributions from these features were considered measures of the local feature’s importance. The final scaled feature importance was obtained by normalizing the feature importance scores by the highest (first-ranked) feature, which was assigned a value of 100. The feature importance scores were produced by averaging 10 repeated feature rankings. The results are presented in Figure 7. Each feature was denoted by its wavelength followed by the feature index number (e.g., 709F001). The highly ranked features had the highest average contributions. The error bar shows the standard deviation of 10 measurements. Interestingly, it was discovered that for some organisms, only specific features (specific moment invariants) significantly contributed to classification. For example, in Figure 7B, the 10th moment invariant from each wavelength contributed the most. In contrast, Figure 7C,E show that each wavelength’s 1st and 2nd moment invariants contributed the most to classification decisions. The 1st moment was identified as the primary contributor to the decision in Figure 7G. The first-ranked features were consistently selected as the most critical contributing feature in every repetition shown in Figure 7A,C,G,H. These features had a score of 100 in every test, indicating they were the most powerful features in determining the presence of particular organisms.

Unlike black-box classifiers such as neural networks, the ENET provides the regression model’s coefficients to determine the globally most important predictive features. Figure 8 depicts the global feature importance for each organism. The scaled feature importance was computed using the absolute value of the coefficients in the ENET model. They were also normalized by the most important (first-ranked) feature on a scale of 0 to 100 and averaged over 10 independently trained models. Results similar to the outcome indicated by LIME were observed and are summarized in Figure 8B,C,E: the 1st, 2nd, and 10th moment invariants were highly ranked. Figure 8F shows that there was no dominant feature allowing the classification of 526—*Arthrobacter*, as the differences among the selected features were not statistically significant. This was also indicated by the LIME result, but not to the extent demonstrated in Figure 8F. The results imply that in the case of some organisms, one cannot pinpoint a single specific feature or wavelength that is responsible for the classification. The unique result for 526—*Arthrobacter* is not a surprise: the scattering patterns looked perceptually very different from the results acquired from other organisms. As illustrated in Figure 7 and Figure 8, (A) and (H) share a common characteristic in that 709F010 and 473F003 had the highest average contribution and importance score.

## 4. Discussion

### 4.1. Motivation for Developing the Hyperspectral ELS System

The ELS method has yielded encouraging results for distinguishing bacterial colonies based on their optical properties. As the phylogenetical distance between bacterial species decreases, however, the morphological characteristics of the colonies become less different and more similar. Classification of these organisms using ELS patterns is quite difficult [9]. Previously studied and inaccurately classified samples were reanalyzed with a hyperspectral instrument to demonstrate potential improvement. The sample set consisted of multiple strains of *Arthrobacter* sp., *Curtobacterium* sp., *Massilia* sp., and *Microbacterium* sp. These organisms were previously investigated by the researchers in the Department of Botany and Plant Pathology at Purdue University using a commercial single-wavelength ELS instrument (BEAM) developed by Hettich Lab Technology (Tuttlingen, Germany). The classification showed modest performance, with the average PPV around 84.2%. For those experiments, the scattering patterns were collected at fixed incubation times of 24, 36, or 72 h, regardless of the colony size. PZM with polynomial order of five was extracted from the patterns, and then utilized to build a linear SVM classifier. The limited accuracy of the classification was also evident when single-wavelength measurement was performed using HESPI to differentiate the sample group. In contrast to the previous experiment using BEAM, where the colonies were of varying sizes, the single-wavelength measurements using a 635 nm band from the SC laser were repeated in this study with colonies of 1 mm diameter. This was done to ensure consistency with hyperspectral readouts since the HESPI used in this study was optimized for colonies of 1 mm. The repeated single-wavelength classification resulted in an average PPV of 82.7% for the linear SVM classifier constructed with PZM with polynomial order of five. Four out of eight classified organisms had PPVs below 85%. The previous experiments that were conducted utilized only a single wavelength, which was a limiting factor in terms of the amount of information that could be extracted from the data. As a result, the classification performance was not optimal for the particular sample group. Additionally, the number of features used in the previous experiments was not optimized for the best possible classification outcome. However, when the polynomial order for PZM was increased to 10 to include more descriptive features, the average PPVs for BEAM and single-wavelength measurements using HESPI improved to 89.1% and 92.0%, respectively. This suggests that the poor performance of the previous experiments was due to a lack of descriptive features, and that even adding more information through the implementation of a hyperspectral approach eventually improved the overall PPV to around 96.4% with a linear SVM model. The scattering patterns measured with the two different single-wavelength measurements and the corresponding classification results using only PZM order of five are presented in the Appendix A.

### 4.2. Prototype Design

Essential to the formation of the scattering pattern was the coherence of the light source. Since the ELS pattern is a manifestation of diffraction phenomena caused by the constructive and destructive interference of coherent light waves, the coherence parameter of the incident light is a crucial factor for correctly creating a distinguishable ELS pattern. Three different light sources were utilized to generate the scattering pattern from *E. coli* K12 colonies to explore the effect of coherence properties. The effect of the white LED flashlight, a 635 nm diode laser, and a 635 nm beam from HESPI are presented in Appendix A. The LED flashlight is an example of an incoherent light source. In this case, diffraction was not created, and only the shadow of the bacterial colony was observed. On the other hand, the diode laser, which emits spatially and temporally coherent light, could easily create visible diffraction patterns. Similarly, the patterns were also observed when HESPI was used as the incident light source. Based on the comparison, we can infer that the combination of the SC laser and AOTF produced a laser beam with sufficient coherent properties.

The attractive property of the AOTF was that it filters a specific wavelength of interest using the acoustic signal and spatially separates the filtered beam at a certain diffraction angle. However, this spatial filtering mechanism caused a design problem during the instrumentation causing the beam spot of the laser to move due to the wavelength-dependent diffraction angle. It was challenging to align the colony with the laser beam unless it moved in the same direction as the beam spot movement. At the imaging plane where the detector was located, the total distance of the beam movement was around 2 mm while the wavelength was shifting from 455 to 715 nm. Nevertheless, the diffraction angle was consistent with the wavelength, so the movement was only in one direction and the displacement was also consistent.

A linear stage with a motorized actuator (Z812B, Thorlabs Inc., Newton, NJ, USA) was integrated into the system to move the AOTF crystal block based on the position of the beam spot in order to compensate for the movement of the beam spot. Appendix A shows the schematic and the picture of the HESPI system with the linear stage. The AOTF crystal block was placed on top of the 1D linear stage operated by a motorized actuator. To compensate for the misalignment, the actuator moved the AOTF crystal block in the opposite direction from the beam spot. The motorized stage was controlled by the PC through a USB connector. Since the location of the beam spot on the sample was consistent with the corresponding wavelength, the displacement of the linear stage was preset in the software, and no feedback control was required. HESPI’s motorized stage resolved the issue with beam spot movement but slowed measurement speed. It took about 10 s to capture 16 spectral patterns. However, the stage’s travel speed (1.6 s/wavelength) was significantly slower than the AOTF’s wavelength shifting speed (0.01 s/wavelength), which negated the greatest advantage of AOTF, which is the rapid wavelength switching.

An alternative optical method was implemented to minimize the movement of the beam spot. The sample and the AOTF were separated by a pair of identical biconvex lenses with 2 inch diameters and 60 mm focal lengths. An optical simulation showing the effect of the two biconvex lenses is illustrated in Appendix A. Two collimated beams, represented in different colors, were exiting the AOTF at different angles and were eventually merged at the sample location. The positions of the lenses determined in the simulation were also experimentally verified. Replacing the motorized stage with the optical component brought several benefits to the instrument. Since the measurement speed was no longer reliant on the linear stage’s travel speed (0.4 s/wavelength), it was significantly increased. By incorporating a camera with a higher frame rate, the measurement speed can be increased further. Moreover, the optical solution was cheaper, simpler, and independent from the control unit. However, the additional optics caused deformation of the beam spot and a slight beam divergence at the imaging plane. Beam distortion was observed at the detector because the diffracted beam passed through lenses with short focal lengths and was severely refracted. Nonetheless, no significant changes were observed in terms of the light-scatter pattern. Since most distinguishable phenotypic features are present in the outer region of the scattering pattern, slight distortion in the beam shape did not pose a significant issue [7].

### 4.3. Classification Result

Utilizing multiple-wavelength or single-wavelength patterns, the hyperspectral method suggested two distinct classification strategies. Comparing the two methods revealed that using more wavelengths significantly improved classification accuracy, as there were many more informative characteristics to describe the colonies. This result suggested that the multiple wavelength method was a more appropriate technique to differentiate the bacteria in general. Nevertheless, despite a lower overall classification performance, the single wavelength method demonstrated that certain colonies can be classified with greater sensitivity using specific wavelengths. This encourages a strategy of selecting optimal wavelengths for the particular type of organism. Adopting a discrete diode laser-based system is more cost-effective, so using an optimized choice of wavelength could reduce the cost of the instrument while maintaining classification performance. Owing to its extensive range and diverse selection of wavelengths, the HESPI can also be used as a reference instrument to determine the single or multiple-wavelength version for a specific application, as it can provide input for the design of a stand-alone system. Regarding future research, due to the robust performance of the hyperspectral approach in classifying the tested bacteria, it is worthwhile to further investigate its effectiveness with bacterial groups at lower taxonomic levels such as serovar or strain level using the hyperspectral system. This would provide valuable insights into the potential of the hyperspectral approach for improving bacterial classification at the subspecies level, with practical applications in fields such as microbiology, food safety, and medical diagnostics.

### 4.4. Feature Selection and Importance

LIME and the elastic net regression evaluated the features by their contributions to the classification model. A direct comparison of the results of the two methods revealed that certain features (particular orders of PZM) participated in predictions more frequently than others. For example, as shown in Figure 7B and Figure 8B, the 10th order PZM had the highest contribution or importance values according to LIME and the elastic net analysis. Similar behaviors were observed in (C), (E), and (H), showing that most of the highly ranked features were similar in both methods. The feature analysis using LIME and elastic net revealed that the classification model relied on specific orders of the PZM rather than a group of features derived from patterns collected at the same wavelength. Although 709 nm had the largest number of features that were highly ranked, this does not necessarily imply that 709 nm was the best-performing wavelength for this application. This is because features obtained from different wavelengths were highly correlated, indicating that every hyperspectral pattern contained meaningful information. The feature correlation heatmap is presented in Figure 9. For example, the feature associated with 1st order PZM in wavelength had absolute correlation values close to 1. The 2nd order PZMs were correlated at 0.8 level.

## 5. Conclusions

This report presents the design and evaluation of a hyperspectral elastic light-scatter phenotyping instrument (HESPI). A proof-of-concept experiment was conducted to test the classification ability of the hyperspectral ELS technique with eight bacteria collected from lettuce leaves. HESPI was able to generate the hyperspectral ELS patterns in 70 different wavelengths in less than 30 s. The varied selection of wavelengths made it possible to generate scattering patterns in the wavelength that cannot be produced by diode lasers currently on the market. The increased number of wavelengths improved classification, resulting in an almost 95% accuracy rate. PZMs were extracted from the collected patterns and served as descriptive and predictive features. The feature selection techniques, a univariate (ANOVA) and multivariate (elastic net regularization), were applied to avoid overfitting. In addition, a post hoc feature analysis was implemented to understand the decisions made by the classifier. LIME algorithm was used to perform local feature analysis. The global feature importance was determined using the elastic net regularized regression coefficients. Despite its strong performance, the HESPI system might benefit from enhancements in multiple areas. Its signal acquisition hardware could be improved to speed up the measurement. Integration of HESPI with another modality, such as vibrational spectroscopy, would be useful in characterizing additional phenotypic traits of the bacterial colonies, resulting in even more precise classification.

## Figures and Tables

**Figure 1 sensors-23-03485-f001:**
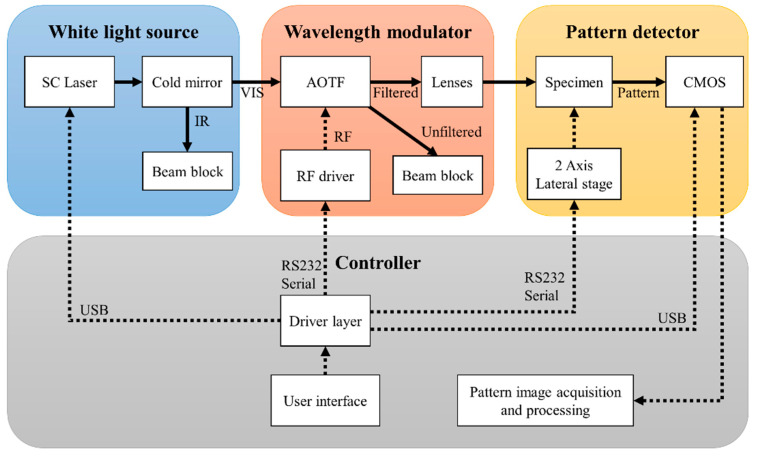
A schematic diagram of the hyperspectral elastic light-scatter phenotyping instrument (HESPI) and flow chart of the measurement. The main components of the instrument are illustrated in different colors. The solid line represents the beam path whereas the dotted line is the sequence of controls.

**Figure 2 sensors-23-03485-f002:**
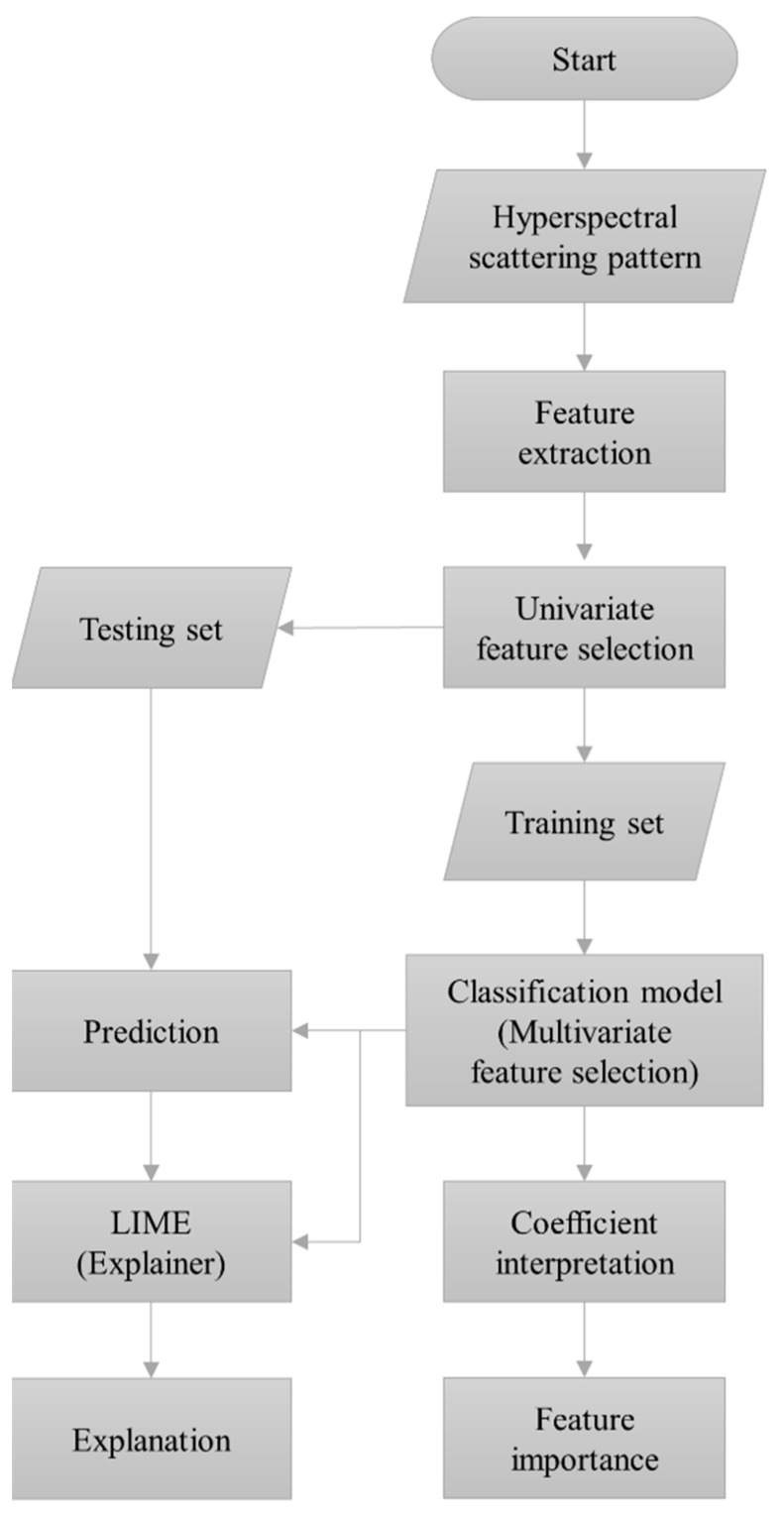
A flowchart illustrating the feature processing methodology for the classification of hyperspectral ELS patterns. Univariate and multivariate feature selection techniques were utilized to alleviate the burden of high feature dimensionality, which increases the computational cost of classification and increases the likelihood of overfitting.

**Figure 3 sensors-23-03485-f003:**
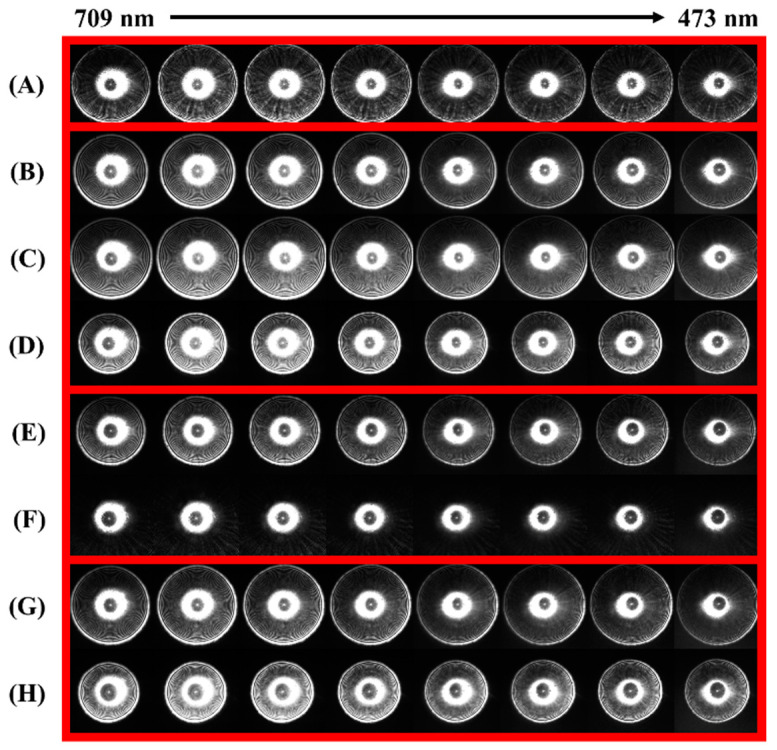
Representative images of the hyperspectral ELS patterns of the eight bacterial colonies: (**A**) 128—*Massilia*, (**B**) 284—*Microbacterium*, (**C**) 410—*Microbacterium*, (**D**) 441—*Microbacterium*, (**E**) 510—*Arthrobacter*, (**F**) 526—*Arthrobacter*, (**G**) 536—*Curtobacterium*, and (**H**) 586—*Curtobacterium*. The numbers before the bacteria name indicate the strain code that was assigned during the previous investigation. The wavelength ranged from 473 to 709 nm, and the sample organisms were grouped by their genera.

**Figure 4 sensors-23-03485-f004:**
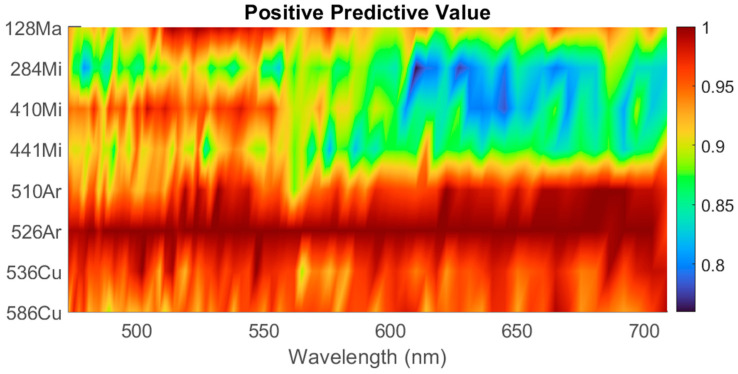
Positive predictive values for each organism in relation to the laser’s incident wavelength. The heatmap demonstrates that certain organisms are distinguished more effectively at specific wavelengths.

**Figure 5 sensors-23-03485-f005:**
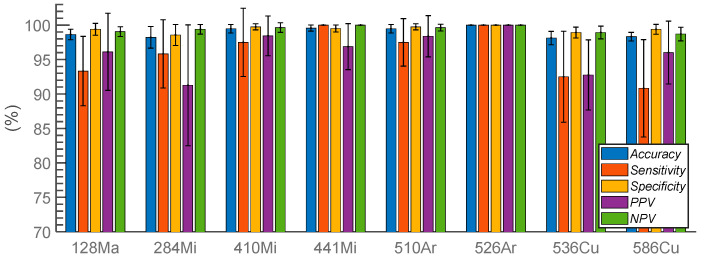
Classification performance of eight bacterial species utilizing the elastic net logistic regression classifier created with hyperspectral ELS data (n = 10).

**Figure 6 sensors-23-03485-f006:**
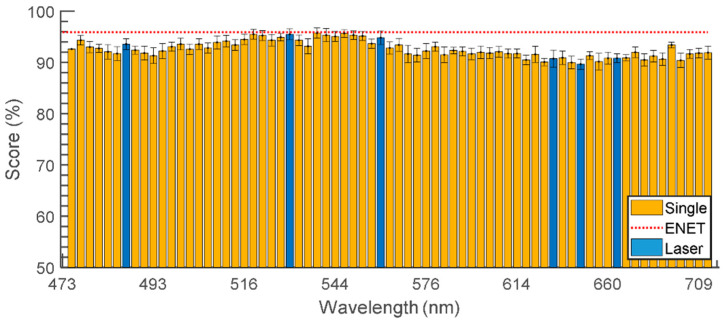
Individual classification scores of SVM-based classifiers using single-wavelength ELS patterns where the bars highlighted in blue represent the common wavelength of diode lasers. The error bars represent the standard deviation. The dotted red line represents the classification score of the elastic net logistic regression model.

**Figure 7 sensors-23-03485-f007:**
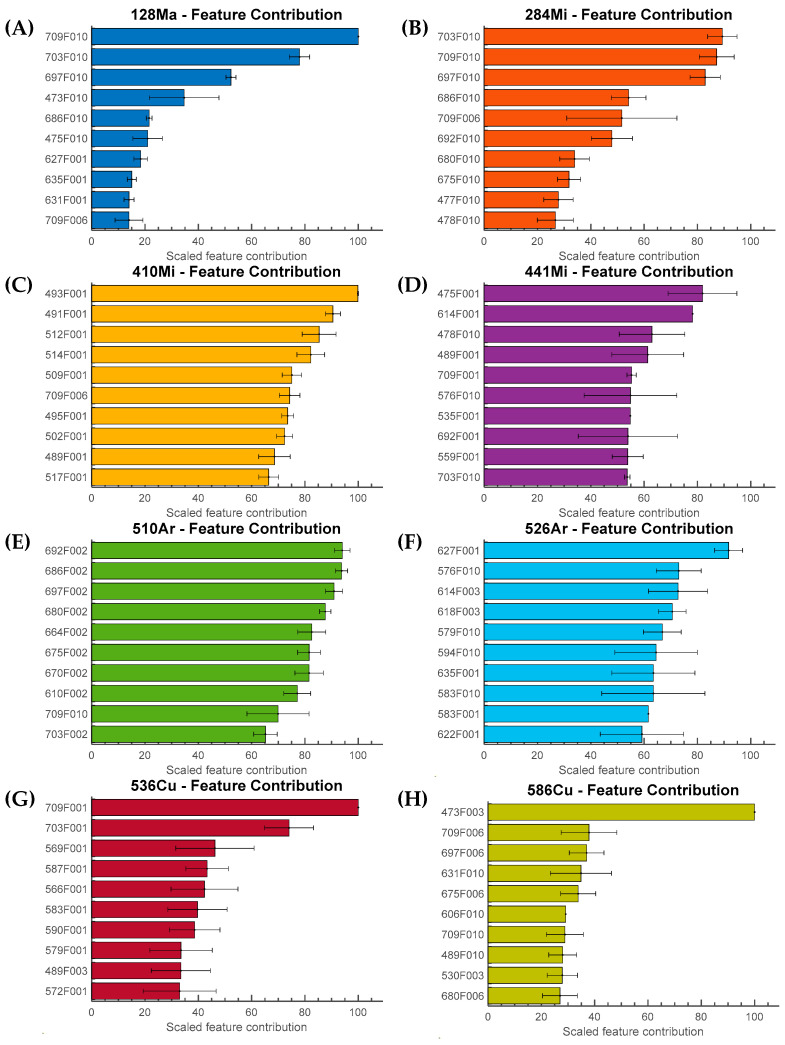
Top 10 contributing features and the normalized contribution values for individual organisms are presented: (**A**) 128-*Massilia*, (**B**) 284-*Microbacterium*, (**C**) 410-*Microbacterium*, (**D**) 441-*Microbacterium*, (**E**) 510-*Arthrobacter*, (**F**) 526-*Arthrobacter*, (**G**) 536-*Curtobacterium*, and (**H**) 586-*Curtobacterium*. The most contributing features for each organism are identified by LIME. The feature contribution was scaled based on the feature with the highest contribution value. The result is an average of 10 repetitions, and the error bar represents the standard deviation. The feature names are defined by their wavelength followed by “F” and the index position of the corresponding feature.

**Figure 8 sensors-23-03485-f008:**
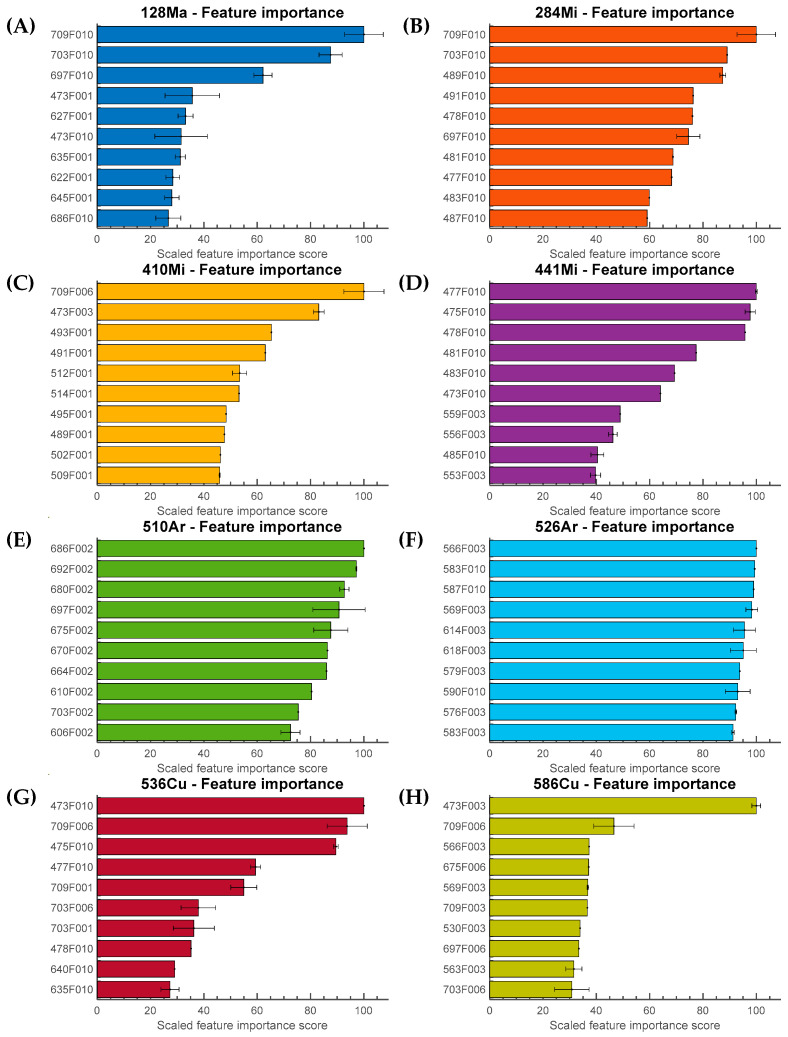
Top 10 predictive features and the normalized feature importance values for individual organisms are presented: (**A**) 128-*Massilia*, (**B**) 284-*Microbacterium*, (**C**) 410-*Microbacterium*, (**D**) 441-*Microbacterium*, (**E**) 510-*Arthrobacter*, (**F**) 526-*Arthrobacter*, (**G**) 536-*Curtobacterium*, and (**H**) 586-*Curtobacterium*. The most predictive features for each organism were identified by the ENET regression model coefficients. The feature importance was scaled based on the feature with the highest absolute coefficient value. The result is an average of 10 repetitions, and the error bar represents the standard deviation. The feature names are defined by their wavelength followed by ”F” and the index position of the corresponding feature.

**Figure 9 sensors-23-03485-f009:**
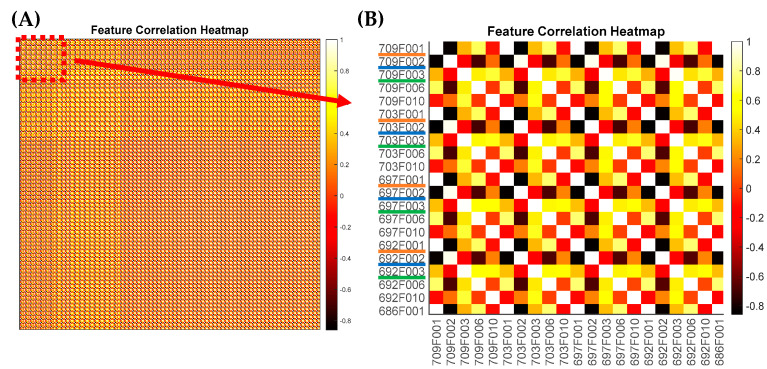
A feature correlation heatmap is presented to demonstrate the correlation between the features across the wavelength: (**A**) shows the overall feature correlation heatmap, whereas (**B**) is the magnified area of the map closely viewing the correlation. The correlation value is between 1 and −1.

**Table 1 sensors-23-03485-t001:** A comparison of three different classification methods by PPVs: individual classifications with a linear SVM for every wavelength, a linear SVM with all features from entire wavelengths, and the elastic net logistic regression model (n = 10).

	128Ma	284Mi	410Mi	441Mi	510Ar	526Ar	536Cu	586Cu
SVM	93.73	85.64	90.46	88.63	95.80	99.87	96.15	94.62
Single wavelength	(3.4)	(3.7)	(5.8)	(3.2)	(2.7)	(0.5)	(2.0)	(2.0)
SVM	93.76	95.71	94.74	94.94	100	100	97.22	94.81
Entire feature set	(0)	(3.9)	(3.0)	(6.4)	(0)	(0)	(3.5)	(2.9)
ENET	96.13	91.27	98.44	96.88	98.39	100	92.77	96.02
Entire feature set	(5.6)	(8.8)	(3.0)	(3.3)	(3.0)	(0)	(5.1)	(4.6)

## Data Availability

The data presented in this study are available on request from the corresponding author, subject to a confidentiality agreement.

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
