# Peer review of "Bacterial Colony Phenotyping with Hyperspectral Elastic Light Scattering Patterns"

_sensors, 2023, doi:10.3390/s23073485_

Round 1

Reviewer 1 Report

In this paper, authors designed and validated a hyperspectral elastic light scatter phenotyping instrument (HESPI) capable of classifying microorganisms, to test the classification ability of the hyperspectral ELS technique. The device is capable of generating hyperspectral ELS patterns of 70 different wavelengths in less than 30 seconds. Increasing the number of wavelengths improved classification and used feature reduction and selection algorithms to enhance robustness and ultimately reduce the complexity of data collection. Feature selection techniques, a univariate (ANOVA) and elastic net regularization are used to avoid overfitting to determine the global feature importance. But there are still problems to be aware of.

1. In the article, the authors mentioned supplementary pictures many times, but I didn't see pictures whose names start with S. I suggest the authors to add them, after all, they are important information for the article content. In addition, for some accuracy results, it is suggested that the authors indicate some excellent specific values, such as Figure 7 and 8, which can give readers more intuitive results.

2. The authors explained in Section 2.2 that the reason for selecting these 8 colonies was the poor classification effect previously. I am curious about the reasons for the poor classification effect, what methods were used before to lead to such results, and the comparison between the previous methods and the proposed HESPI.

3. At wavelengths in the 473-709 nm region, the accuracy of the classification results is between 88.7% and 93.2%. This accuracy still seems to be not very high. I am curious about the authors' detailed thinking on the causes of this condition.

4. The whole structure of this paper is incomplete. The future work is missing at the end of this paper, it should be supplied to further demonstrate the application prospect, improvement direction or practical significance of your proposed methods.

5. Feature extraction and correlation processing play an important role in the authors' work, their applications also show up in many ways, such as PM2.5 feature recognition and air pollution control. It is suggested that the authors cite the following papers in related work, to illustrate the practicability and extensiveness of feature extraction to prove the superiority of the method chosen by the authors: “Ensemble meta-learning for few-shot soot density recognition”, “PM2.5 monitoring: Use information abundance measurement and wide and deep learning”, “Deep dual-channel neural network for image-based smoke detection”.

Reviewer 2 Report

The authors address a hyperspectral detection and classification of bacterial colonies. For this application, they propose supercontinuum coupled with acousto-optical  filter. This type of tunable illumination is well-known and has been already used for hyperspectral imaging though its application to bacteries might be interesting to biophotonics community. To fit the scope of Sensors, I recommend to enhance the technical/engineering side of the paper and suggest the following revisions.

1. Acousto-optical interaction is a flexible technique widely spread in hyperspectral imaging including microscopy and biomedical applications. Please expand Introduction by thorough overview of existing AOTF-based biological setups and emphasize the novelty of your instrument.

2. Please describe your setup in more detail. What AOTF do you use? Please either specify its model or present its parameters (cut angle, spectral resolution, angular aperture, etc.).

What scheme of acousto-optic diffraction do you use and why? Image quality, resolution and other parameters depend greatly on how you install AOTF ( https://www.mdpi.com/1996-1944/14/11/2984).

For the presented spectral images, please specify gain, exposure time, binning and other parameters of the camera.

How did you calibrate the setup to get meaningful spectral dependencies? How do you cope with image aberrations (https://doi.org/10.1364/JOSAA.34.001109), vignetting and other issues? With regard to hyperspectral imaging modality, radiometric and spatio-spectral calibration are an important stage in the measurement pipeline (https://doi.org/10.1016/j.chemolab.2009.11.012, https://doi.org/10.1117/12.2541049, https://doi.org/10.1134/S0020441216040217). Figure 1 should be modified with respect to setup calibration, data correction and image matching procedures.

3. Classification algorithm is based on the spectral descrepancies between the bacterial colonies. Please show the spectra measured in a few points of the images presented in Figure 3.

Reviewer 3 Report

In this paper, authors reported the development and validation of a hyperspectral ELS phenotyping instrument (HESPI), consisted of a supercontinuum (SC) laser and an acousto-optic tunable filter (AOTF). The classification ability of the hyperspectral ELS system was evaluated using 8 different bacteria species. However, this method has been reported in a conference paper ( Proceedings Volume 12120, Sensing for Agriculture and Food Quality and Safety, 1212005 (2022) https://doi.org/10.1117/12.2623267). Besides, the data processing methods are also common. Thus, the paper is not novel enough to publish in Sensor journal, unless the authors fully explain its innovation. Some other revised suggestion were shown below.

1.   More details should be added to show how the intensity values of captured patterns were adjusted based on a reference pattern? The necessary formula might be better.

2.  In lines 238-240 “Previous studies on the wavelength-dependent ELS pattern characteristics predicted and explicitly explained this phenomenon”. The authors should give some summarized explanation directly.

3.   In lines 250-252 “PZMs were extracted from each pattern using the 10th-order pseudo-Zernike polynomials, which yielded the most accurate classification results demonstrated after an iterative search.” This conclusion is not convincing, since the changed values for PZ order is not enough.

4.   The five evaluation indexes in Figure 5 should be defined.

5.  The authors should make a comparison with other reported literatures, especially for hyperspectral and multispectral approaches.

6.   Most of cited references are too old, and more literatures published in recent 5 years should be added.

Reviewer 4 Report

The manuscript presents a methodology for a rapid identification of pathogens presents on leafy vegetables. It is highly technical and versatile. Manuscript is written well and not difficult to understand by an average reader.

There are few questions,

1. Are the measured spectral features  of the  properties the colony or the individual cell?

Will you be able to classify the pathogen as a single cell on the respective green leaf? Can you spike a leaf with a pathogen and identify without growth?

2. What is the source of these pathogens? Are they available to anyone interested?

3. What is spectral collection time? How long does it to take a measurement? 

4. How long does it take from sample collection to identification without the growth?

5. Explain why scattering patterns of 526-Ar- 233 Throbacter is different form others.

6. List possible reasons why certain wavelengths are not helpful in predictive probability.

7. Only about 3% of the data was chosen for classification and that is disappointing. The selection has no objectivity. Can you come to a objective conclusion about the right amount data to be used through a calculation/classification? 

Round 2

Reviewer 2 Report

The authors did a good job and addressed all my concerns. In the present form, the manuscript together with supplementary document looks fine for publication. 

Reviewer 3 Report

Since most of the suggestions were answered well, the revised manuscript is now ready to be published.